# Toolkit for Population Health Initiatives Around the Globe Related to Collaborative Comprehensive Medication Management for Children and Youth

**DOI:** 10.3390/children6040057

**Published:** 2019-04-08

**Authors:** Richard H. Parrish, Johannes van den Anker

**Affiliations:** 1Department of Pharmacy Services, St. Christopher’s Hospital for Children—American Academic Health System, 160 E. Erie Avenue, Philadelphia, PA 19134, USA; 2Virginia Commonwealth University, School of Pharmacy, Richmond, VA 23298, USA; 3Universitäts-Kinderspital beider Basel (UKBB), Spitalstrasse 33, CH-4031 Basel, Switzerland; JohannesN.VandenAnker@ukbb.ch; 4Children’s National Health System, 111 Michigan Avenue, Washington, DC 20010, USA; 5Erasmus Medical Center—Sophia Children’s Hospital, s-Gravendijkwal 230, 3015 CE Rotterdam, The Netherlands

**Keywords:** clinical pharmacy, pediatrics, collaboration, population health, comprehensive medication management, children, global

## Abstract

Almost 30 million babies worldwide are born prematurely or become ill annually and need specialized care to survive. Formalized collaborative practice agreements (CPA) between clinical pharmacists and physicians have been put forward as a means for improving the overall medicating experience in many patient populations, including children. This report briefly describes opportunities for collaboration using examples from countries on each continent where CPA is established in professional governance documents and standards. It also provides resources in the form of a toolkit for countries and pharmacist–physician collaborators to authorize and form CPAs to provide comprehensive medication management (CMM) for children and youth with special health care needs (CSHCN).

## 1. Introduction

Almost 30 million babies worldwide are born prematurely or become ill annually and need specialized care to survive [1]. The Convention on the Rights of the Child was a clarion call to rethink how care is delivered in hospital and ambulatory environments to the most vulnerable and marginalized children around the world [2]. Unfortunately, one-third of the world’s population lack regular access to essential medication [3]. In addition, it has been estimated that $500 billion annually could be saved through the responsible use of medication [4]. Moreover, as early as 2006, the World Health Organization (WHO) and the International Pharmaceutical Federation (FIP) recognized the paradigm shift occurring worldwide in the practice of pharmacy and developed a handbook for practitioners, educators, researchers, and policy makers to enable meaningful changes to the scope of the practice [5]. A 2009 FIP statement on collaborative pharmacy practice identified five levels of collaboration, from minimal contact with other health professionals to the authority to initiate or modify pharmacotherapy [6]. The importance of having adequate pharmacy technicians and other pharmacy support staff to allow clinical pharmacists to engage in advanced practice activities cannot be overemphasized [7]. The purpose of this paper is to provide a toolkit for global organizations in the developed and developing world to provide collaborative care (CPA) and comprehensive medication management (CMM) for children and youth with special health care needs (CSHCN). To provide specific resources for countries to initiate and sustain collaborative practices and/or networks for providing children’s care, this paper has sections focused on different continents.

To estimate whether a particular country has the capacity for CMM through CPA, an internet search was performed using the following search terms: children, pediatrics, clinical pharmacy, collaborative practice, collaborative care, pharmacist prescribing (PhP), partnership for care, board of pharmacy, pharmacy council, pharmacy guild, pharmaceutical society, regulatory, medication review, medication management, medication access, scope of practice, and each continent. While an Internet search may be limited as far as creating a comprehensive survey of practice expectations worldwide, the authors realized that a search of this nature may not capture all possible activity in other countries. In addition, the report was limited to English language websites and those that could be translated. The toolkit provides tools and resources for potential collaborators to assess the value of closer collaboration in practice for the benefit of patients, including children. Moreover, the expansion of clinical pharmacy education within universities may be an indicator of an evolving societal expectation for collaborative CMM practice. Readers are referred to Appendix B for further information about educational standards, especially Fathelrahman et al. The jurisdictional or regulatory documents of the countries listed in this report have a reference to CPA, CMM, or PhP and are profiled in World Health Organization (https://www.who.int/maternal_child_adolescent/child/en/), UNICEF (https://www.unicefusa.org/), International Pharmaceutical Federation (FIP; http://www.fip.nl/), or Commonwealth Fund (https://international.commonwealthfund.org/) publications. Countries listed in this report serve as examples of the scope of regulatory and practice scenarios and processes that could be applied in the developing world. Appendix A provides a continent comparison of the extent to which key clinical pharmacist activities are expected in daily practice [8].

## 2. Africa

In South Africa, pharmacists are registered with the South African Pharmacy Council. While pharmacists in hospital practice may contribute to antimicrobial stewardship programs (e.g., switches from IV to oral administration), provide pharmacokinetic advice (e.g., therapeutic drug monitoring), or service anticoagulant clinics, they are unable to alter doses or order laboratory tests on their own [9]. Domain 5.7 of the 2018 Competency Standards for Pharmacists includes a behavioral statement on collaborative practice in terms of the entry level to practice, intermediate practice, and advanced practice [10,11].

Useful links on collaborative practice and regulation:South African Pharmacy Council (SAPC) (https://www.pharmcouncil.co.za/)Pharmaceutical Society of South Africa (https://www.pssa.org.za/)South African Society of Clinical Pharmacy (https://www.sasocp.co.za/)

## 3. Asia

In Singapore, registration for specialist pharmacists includes advanced pharmacotherapy for cardiology, geriatrics, infectious diseases, and psychiatry. Oncology pharmacy, critical care pharmacy, and pediatric pharmacy are also recognized by the Pharmacy Specialists Accreditation Board. Board certifications from the Board of Pharmacy Specialties (BPS) and case write ups and logs are required for advanced practice [12].

Useful links on collaborative practice and regulation:Health Sciences Authority (Registration of Pharmacies; http://www.hsa.gov.sg/content/hsa/en/Health_Products_Regulation/Manufacturing_Importation_Distribution/Overview/Audit_Licensing_and_Registration_Of_Pharmacies.html)Singapore Pharmacy Council (http://www.healthprofessionals.gov.sg/spc)Therapeutics Products Division (http://www.hsa.gov.sg/content/hsa/en/Health_Products_Regulation/Western_Medicines/Overview.html)Pharmaceutical Society of Singapore (PSS; http://www.pss.org.sg/)Pharmacy Specialists Accreditation Board (http://www.healthprofessionals.gov.sg/psab)

## 4. Europe

In the Netherlands, the Charter Professionalism of the Pharmacist governs the ethical and practice standards for collaborating with other health care professionals [13]. The professional context of pharmacists includes collaboration, referred to as the complementarity of pharmacists and physicians. “Physicians and pharmacists play the following roles in the care process: the physician makes a diagnosis and, after discussing the options with the patient, recommends a choice of treatment. The pharmacist then advises on the translation of the treatment into a suitable pharmacotherapy product for the patient”. Clearly, the Dutch Minister of Health, Welfare, and Sport and the House of Representatives support “close collaboration” with general practitioners and specialists for advise on medication optimization [14]. A systematic review of non-dispensing pharmacists embedded in primary care practice revealed that fully integrated pharmacists can improve patient-centered care [15]. Recently, the Dutch Pediatric Pharmacotherapy Expertise Network or Nederlands Kenniscentrum voor Farmacotherapie bij Kinderen (NKFK) developed an evidence-based framework for dosing children’s medication and combined it with the Dutch Pediatric Formulary [16].

Useful links on collaborative practice and regulation:The Royal Dutch Pharmacists Association (https://www.knmp.nl/knmp/about-knmp)Kinderformularium (https://www.kinderformularium.nl/)

In the United Kingdom, Standard 2 of the standards for pharmacy professionals of the General Pharmaceutical Council (GPhC) states that pharmacy professionals work in partnership or collaboration with patients and others on the health care team [17]. Just this year, GPhC released standards for the education and training of pharmacist-independent prescribers [18]. In this document, all non-medical prescribers take responsibility for the clinical assessment of the patient; establishing a working diagnosis; and specifying the clinical management required, as well as the responsibility and appropriateness of any prescribing [17]. Medication optimization is a major trend to ensure safer prescribing that is based on shared decision-making, medication review and reconciliation, and optimization of key therapeutic areas (such as antimicrobials) [19].

Useful links for collaborative practice and regulation:Medical and Healthcare Products Regulatory Agency (MHRA; https://www.gov.uk/government/organisations/medicines-and-healthcare-products-regulatory-agency)The European Medicines Agency (http://www.emea.eu/)General Pharmaceutical Council (http://www.pharmacyregulation.org/registration/registration-pharmacy-premises)Royal Pharmaceutical Society of Great Britain (http://www.rpharms.com/home/home.asp)Pharmaceutical Group of the European Union (https://www.pgeu.eu/en/pgeu/members.html)British National Formulary for Children (https://about.medicinescomplete.com/publication/british-national-formulary-for-children/)

## 5. Oceania

In New Zealand, the Pharmacy Council registers pharmacist prescribers that work in a collaborative health team environment and are authorized to initiate, modify, maintain, or discontinue prescription therapy; order and interpret investigations; assess and monitor a patient’s response to therapy; and provide education and advice to a patient regarding their therapy [20]. Practice area examples include specializing in a therapeutic area, generalist practice in primary care or the emergency department, high risk areas of the continuum of care, and specific classes of medications [21].

Useful links on collaborative practice and regulation:Medsafe (http://www.medsafe.govt.nz/)Pharmaceutical Society of New Zealand (http://www.psnz.org.nz/)Pharmacy Council (http://www.pharmacycouncil.org.nz/index.asp)Pharmacy Guild of New Zealand (http://www.pgnz.org.nz/)

In Australia, Standard 9 of the 2017 Professional Practice Standards specifically lists collaborative care, the responsibilities of pharmacists in general practice clinics, the provision of Aboriginal and Torres Strait Islander people, and practice in residential care facilities and medical home practices. Pharmacists also have a responsibility to provide health screenings and risk assessments (Standard 10) for estimating cardiovascular, diabetes, obstructive sleep apnea, and skin cancer risk using standardized tools (see links below). Pharmacists also provide vaccinations (Standard 11), minor ailments service (Standard 12), disease state management (Standard 13), medication review (Standard 14), dose administration aid (Standard 15), and harm minimization (Standard 16) in a collaborative fashion [22].

Useful links on collaborative practice and regulation:Therapeutic Goods Administration, Department of Health and Aging (http://www.tga.gov.au/)The Pharmacy Guild of Australia (http://www.guild.org.au/)The Pharmaceutical Society of Australia (http://www.psa.org.au/)International Health Care System Profiles-Australia (https://international.commonwealthfund.org/countries/australia/)Clinical Information Access Portal of New South Wales (login required; https://www.ciap.health.nsw.gov.au/browse/paed.html)

## 6. South America

In Brazil, national Resolutions 585/2013 and 586/2013 and Federal Law 13021/2014 changed the classification of pharmacists and established the title of professional specialist for pharmacists in Brazil, which includes the title of specialist in clinical pharmacy [23]. In addition, a clinical services conceptual framework has also been published, and its content modifies what is proposed in the Brazilian Classification of Occupations, particularly with regards to clinical pharmacy [24]. Collaborative practice is in its formative phase with the establishment of the Brazilian Society of Clinical Pharmacy (2017). Pharmacists are independent prescribers for non-prescription medications and dependent prescribers continuously used for refills or therapy previously prescribed by a physician [23]. Like other countries, the lack of mass-produced child-friendly preparations for clinically important products directly affects child health [25].

Useful links on collaborative practice and regulation:Brazilian Society of Pharmacists and Community Pharmacies (https://sbffc.org.br/)Support Program for Pharmaceutical Care in Healthcare (http://www.cff.org.br/userfiles/file/_PROFAR_kit_Livro_corrigido.pdf)National Health Surveillance Agency (http://portal.anvisa.gov.br/)

## 7. Summary

The purpose of this toolkit is to aid countries and clinical pharmacists around the world in identifying opportunities to develop more formalized collaborative practices between clinical pharmacists and other prescribers for managing the medications of children. Progress related to CMM and CPA in the developed world may be applicable in other jurisdictions and settings.

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
