# Peer review of "Toolkit for Population Health Initiatives Around the Globe Related to Collaborative Comprehensive Medication Management for Children and Youth"

_children, 2019, doi:10.3390/children6040057_

Round 1

Reviewer 1 Report

A nice review of the role of the pharmacist on a global nature. Items to consider:

An internet search of the terms described may be limited as an actual interview with a questionnaire of specific clinical pharmacist tasks could be address by an appropriate representative. For example, your data from Asia is probably underrepresented as there is an active group in developing clinical pharmacy practice in China/Japan/Korea.  I would think there is more information for these countries that is missing. Bottom line: your data is probably underreported.

A comparison table of key clinical pharmacy tasks would be useful with the comparison with USA practice standards.

It would be interesting to see if you could actually tease out Pediatric specific practices since there are vast differences within the pharmacy world as to how they are operationalized.  Just looking at the state of practice in the USA,pediatric resources may not be as ample when compared to adult resources in multi-center practice facilities. A properly constructed survey addressing actual practicing Pediatric institutions could serve a surrogate for assessing the state of true Pediatric practice.

Author Response

We appreciate the reviewer's comments and suggestions. While an internet search may be limited as far as a creating a comprehensive survey of practices worldwide, our intention was clearly stated in the introduction that the countries listed in this report serve as examples that could be applied in the developing world. We realize that a search of this nature may not capture all possible activity in other countries. In addition, the report was limited to English language websites and those that could be translated. The purpose of this paper was not to examine the state of pediatric practice relative to collaboration between physicians and pharmacists worldwide. There is wide variability in North America as well. The toolkit provides tools and resources for potential collaborators to assess the value of closer collaboration for the benefit of patients, including children.

We have included a table that defines the clinical activities of pharmacists with an estimation of the expectation to practice in each continent.

Our experience in practicing on other continents indicates that collaborative practice for comprehensive medication management for children's care is nascent. Therefore, a survey to determine the extent to which collaborative practice for comprehensive medication management for children would not generate the data the reviewer would be looking for.

Reviewer 2 Report

The standard way to describe MTM is MEDICATION therapy review.  This manuscript chooses to use medicine in a number of places, including the abstract, when medication would be more in line with contemporary manuscripts.

Great resources provided

Author Response

We appreciate the reviewer's comments, and have changed medicines to medication throughout the paper.

Round 2

Reviewer 1 Report

Please provide a key for the star scoring system on the table you have provided. The added comment on the limitation of your internet search methodology (English/English assessible) is very useful. 

Have you considered expanding your search on clinical pharmacy instruction in countries to serve as a basis for having pharmacists trained in the clinical domain?  Granted that there is no formalized recognition, the individual practitioners may be trained to make clinical interventions.  You will find an abundance of information in the literature. 

Author Response

Please provide a key for the star scoring system on the table you have provided. The added comment on the limitation of your internet search methodology (English/English assessible) is very useful. 

A scoring key was added to the table.

Have you considered expanding your search on clinical pharmacy instruction in countries to serve as a basis for having pharmacists trained in the clinical domain?  Granted that there is no formalized recognition, the individual practitioners may be trained to make clinical interventions.  You will find an abundance of information in the literature. 

In our academic experience, there is a huge gap between what is taught in university and what is practiced. The purpose of the paper is to provide practicing pharmacists with a toolkit that can be applied. While clinical education may be an indicator of an evolving societal expectation for collaborative CMM practice, a review of clinical education within schools of pharmacy around the world was beyond the scope of this paper. A cursory review  can be found in Appendix 1 - Fathelrahman, A.; Ibrahim, M.; Wertheimer, A.I. (eds.). Pharmacy Practice in Developing Countries: Achievements and Challenges. London: Elsevier (2016). Available online: https://www.sciencedirect.com/science/article/pii/B9780128017142020013 

We included a statement about educational evolution and sources in the second introductory paragraph.
